# Evidence of West Nile Virus Circulation in Horses and Dogs in Libya

**DOI:** 10.3390/pathogens13010041

**Published:** 2023-12-31

**Authors:** Kholoud Khalid Ben-Mostafa, Giovanni Savini, Annapia Di Gennaro, Liana Teodori, Alessandra Leone, Federica Monaco, Mohammed Masoud A. Alaoqib, Abdunnabi A. Rayes, Abdunaser Dayhum, Ibrahim Eldaghayes

**Affiliations:** 1Department of Microbiology and Parasitology, Faculty of Veterinary Medicine, University of Tripoli, Tripoli P.O. Box 13662, Libya; 2National Center for Animal Health, Tripoli P.O. Box 83252, Libya; 3Department of Virology and Tissue Culture, Istituto Zooprofilattico Sperimentale dell’Abruzzo e del Molise “G.Caporale”, 64100 Teramo, Italy; 4Department of Internal and Infectious Diseases, Faculty of Veterinary Medicine, Omar Al-Mukhtar University, Albaida P.O. Box 919, Libya; 5Department of Internal Medicine, Faculty of Medicine, University of Tripoli, Tripoli P.O. Box 13932, Libya; 6Department of Preventive Medicine, Faculty of Veterinary Medicine, University of Tripoli, Tripoli P.O. Box 13662, Libya

**Keywords:** West Nile virus, horses, dogs, seroprevalence, Libya

## Abstract

West Nile virus (WNV) is a globally significant mosquito-borne Flavivirus that causes West Nile disease (WND). In Libya, evidence of WNV circulation has been reported in humans but never in animals. The aim of this study was to determine the seroprevalence of WNV infection in horses and dogs in Libya. In total, 574 and 63 serum samples were collected from apparently healthy, unvaccinated horses and dogs, respectively, between 2016 and 2019. A commercially available competitive enzyme-linked immunosorbent assay (c-ELISA) kit was initially used to test the collected samples for the presence of WNV Ig-G antibodies. Positive and doubtful sera were also tested using a more specific virus neutralisation assay to confirm whether the ELISA-positive results were due to WNV or other Flavivirus antibodies. The seroprevalence of WNV IgG antibodies according to ELISA was 13.2% out of 574 of total horses’ samples and 30.2% out of 63 of total dogs’ samples. The virus neutralisation test (VNT) confirmed that 10.8% (62/574) and 27% (17/63) were positive for WNV-neutralising titres ranging from 1:10 to 1:640. Univariable analysis using chi-square tests was conducted to measure the statistical significance of the association between the hypothesized risk factors including city, sex, breed, and age group and were then analyzed using the subsequent multivariable logistic regression model for horse samples. Age group was found to be the only significant risk factor in this study. The results of the present study provide new evidence about WNV circulation in Libya.

## 1. Introduction

West Nile virus (WNV) is a single-stranded RNA virus that belongs to the Flavivirus genus of the Flaviviridae family. It was first isolated in 1937 from a febrile patient in the West Nile province of Uganda [1]. Following its first isolation, WNV has spread in Africa, the Middle East, Europe, and America [2,3,4]. It is responsible for neurological symptoms in humans and animals and is currently considered a serious public health problem worldwide, causing outbreaks and fatalities among humans [5].

This virus is maintained in nature in an enzootic cycle involving competent mosquitoes and a wide variety of reservoir host bird species [6,7], which generally act as amplifying hosts and are responsible for the maintenance and spread of infection. As WNV is a vector-borne disease, many environmental factors contribute to its occurrence and emergence, including weather patterns, viral adaptation to local vectors, and bird migration. WNV has been isolated from numerous bird species. In some of them, the infection has been shown to cause specific pathological changes in various tissues, particularly of the central nervous system (CNS) [8].

Various animals, including domestic, companion, and wildlife species, can be infected with WNV [5,9,10,11,12]. When infected, most mammals, including humans and horses, usually act as incidental and dead-end hosts [13]. In other words, due to low levels of viraemia, they are not able to transmit WNV to competent vectors. An exception in that regard is the possibility of WNV transmission among humans through blood transfusion, organ transplantation, or by handling the live virus in a laboratory [5].

WNV replication can occur in various organs and tissues, including lymph nodes, spleen, kidney, muscle, and the central nervous system [14]. Our current understanding of the disease includes its presentation as a neuroinvasive neurologic disease, in which the virus attacks and infects the nervous system as well as other organs [15,16,17].

Following WNV infection, mammals can develop low viraemia titres after subcutaneous or intravenous inoculation with WNV [18]. Dogs and horses can be important sentinels for WNV circulation and potential human exposure—horses because they are very susceptible to WNV infection and dogs because they can provide important information on WNV circulation in urban areas and environments close to humans. They can develop good antibody titre levels that are easy to measure [19].

In humans and horses, most WNV infections are asymptomatic. Clinical manifestations occur very infrequently and can involve the neurological system. West Nile neuroinvasive disease (WNND), West Nile meningitis (WNM), West Nile encephalitis (WNE), and West Nile acute flaccid paralysis, a poliomyelitis-like syndrome known as West Nile poliomyelitis (WNP) [5], have been described in humans; their severity depends on the age and immune status of the patient [20]. In a study in South Africa, 52% of horses positive for WNV had a fever, 92% displayed neurological signs, and 39% died [21].

In other mammalian and nonmammalian species, including dogs, cattle, sheep, goats, camels, deer, squirrels, and reptiles, WNV infection can elicit antibodies [22]. Because they often share a domestic environment with humans, dogs, which can be accidentally infected with WNV, can be important sentinels indirectly indicating viral circulation in urban and suburban areas even before the onset of human disease in the population [23,24].

WNV outbreaks have been observed in many North African countries, including Algeria, Morocco, and Tunisia [24]. Among equids, symptomatic infections and fatalities have been reported in Morocco [25]. However, no information is available about WNV circulation in Libya. The only study on WND seroprevalence in humans in Libya, published in 2017, reported 11 positive samples out of 400 samples tested (2.75%) using enzyme-linked immunosorbent assay (ELISA) [26].

ELISA is the most commonly used diagnostic method for the detection of anti-WNV antibodies in humans and animals [27]. However, the use of ELISA without a more specific WNV neutralisation test could result in false positive results due to the potential of cross-reactivity between closely-related pathogens, such as the Usutu virus (USUV), the St. Louis encephalitis virus, and the Japanese encephalitis virus (JEV).

Even if both species, horses and dogs, are dead-end hosts, WNV infection progresses in different ways. In horses, it can cause death and/or severe clinical signs, whereas in dogs, only serological evidence can confirm the infection. The aim of this work was to acquire current information about the circulation of WNV in Libya through a seroprevalence study carried out on horses and dogs. In addition, we aimed to assess the risk factors associated with WNV seropositivity in these animals. This is the first study of WNV seroprevalence in animals in Libya.

## 2. Materials and Methods

### 2.1. Study Design

A cross-sectional study was conducted between 2016 and 2019 to investigate the serological prevalence of and exposure to WNV in apparently healthy horses and dogs in some of the western and eastern regions of Libya. These areas were selected because they have an ecological environment that is suitable for the life cycle of the virus. Among the eight high-risk “hotspot” locations (cities) in Libya (Tobruk city, Al Marj city, Ajdabiya city, Misrata city, Tripoli city, Al Zawia city, Gharyan city, and Hamada Hamra district) (Figure 1) serosurveillance was conducted in three locations: samples were collected from Al Marj city in eastern Libya and from Al Zawia city and Tripoli city in western Libya (Figure 2). The information during sample collection, as expected risk factors, including city, sex, breed, and age group were collected for horse samples.

The criteria for identifying hotspot areas were as follows:(1)Suitable environment for vectors based on rainfall and green vegetation areas (FAO’s RVF Early Warning/Decision Support Tool).(2)High animal density and frequent movement based on animal movement survey and information available at the National Center for Animal Health (NCAH) in Libya.

Due to the lack of availability of epidemiological data on WND in Libya, the expected prevalence was based on published information from regional and neighbouring countries [9,28,29,30,31]. The proximate of the prevalence was 30% with CI 95%.

### 2.2. Targeted Animals and Sampling Strategy

Samples were collected from dogs and horses of various breeds with no clinical signs related to WNV disease. The owners declared that these animals had spent all of their lives in the areas where sampling took place. Therefore, we were able to assign a specific area to each sample. Each animal was bled once. Local information about the epidemiological status of the sampling areas was obtained from an animal health centre in Tripoli.

To determine the sample size (n) in order to estimate the prevalence of WNV, p (D+), in the horse population, estimates were calculated based on the probable level of disease (p) from the available evidence in regional and neighbouring countries. The estimates indicated that approximately 30% of the horse population would have antibodies to WNV and the survey results were estimated to be within 4% of this level.

So, we had: *p* = 0.3 *q* = 1 − *p* = 0.7 *l* = 0.04
$$n=\frac{4pq}{l^{2}}=\frac{4\times 0.3\times 0.7}{{\left(0.04\right)}^{2}}=525$$

Then, 10% was added to the sample size to reduce any non-response bias (n = 574). An android application named “Statistics and Sample Size Pro” has been used to calculate the sampling size. By providing the same inputs, we obtain similar estimates. https://play.google.com/store/apps/details?id=thaithanhtruc.info.sass&hl=en&gl=US (accessed on 1 November 2023). The proportional sampling [32] was calculated from each selected area based on the number of horses in the area. In the Al Zawia area (Surman city, Al Zawia city, and Zawarah city), 110 samples were collected; in Greater Tripoli (Tripoli, Gasr Ben Ghashir, and Al Swani), the number was 365; and in Al Marj city, 99 samples were collected (Figure 2).

Samples from dogs were collected in Tripoli only, based on published results of WND in humans in Libya [26], so the dog samples were collected from animals that lived in the same area where the positive human cases were reported, under the assumption that dogs can be considered as good sentinels for monitoring WNV.

Data were collected by using a questionnaire in which the most common variables typically associated with WNV infection were considered. Questions were asked about the location, age in years and months, sex, breed, and use of each animal. Information about clinical signs and vaccination strategies within the last 3 months was also collected, as the animals were in good health and had not been vaccinated against WNDV. Data on the breeding system used and the migratory birds present in the area were also obtained.

### 2.3. Sample Data

In total, 574 samples from horses and 63 samples from dogs were collected. The horses ranged in age from 2 to 240 months, and the dogs ranged in age from 3 to 72 months. Males represented 49.3% of the horse group. The horse breeds included in this study were Arabian (n = 145), local thoroughbred (n = 202), imported thoroughbred (n = 93), and local Libyan (n = 35) (Table 1). These horses originated from three Libyan districts including seven cities: Al-Marj (n = 99), Gasr Ben Ghashir (n = 140), Al-Swani (n = 68), Zuwarah (n = 56), Tripoli (n = 157), Al-Zawia (n = 26), and Surman (n = 28). All dog samples were collected in Tripoli. Males represented 49.2% of the dog group.

### 2.4. Collection and Processing of Blood Samples

Each 5 mL blood sample was collected in a plain dry tube through venipuncture of the jugular vein using a sterile needle and syringe directly after clinical examination. The samples were transported at 4 °C to the laboratory for serum separation within 24 h. In the laboratory, they were centrifuged at 3000 rpm for 10 min, and the resulting sera were transferred into two Eppendorf tubes and stored at −20 °C until further use.

### 2.5. Serological Tests

Serological assays were performed following the recommendation of the WOAH Terrestrial Manual 2018. Both competitive ELISA (c-ELISA) and virus neutralisation tests (VNTs) were performed at the WOAH Reference Laboratory for West Nile Disease, Istituto Zooprofilattico Sperimentale “G. Caporale”, Teramo, Italy.

#### 2.5.1. Competitive Enzyme-Linked Immunosorbent Assay (c-ELISA)

Dog and horse serum samples were screened by c-ELISA (ID Screen^®^ West Nile Competition Multi-species; IDvet, Grabels, France) for the presence of IgG antibodies against Flaviviruses. The test is directed against an epitope of the E protein that is common to WNV and other Flaviviruses and proven to detect a wide range of Flavivirus antibodies including WNV, JEV, tick-borne encephalitis virus (TBEV), USUV, Zika virus (ZIKV), and Dengue virus (DENV) in multiple species, including humans, horses, dogs, birds, and others. c-ELISA was performed following the manufacturer’s protocol. The results were interpreted by calculating the OD and the sample/negative control ratio (S/N%) as described in the manufacturer’s guidelines. An S/N ratio less than or equal to 40% indicated a positive serum sample, between 40% and 50% was considered inconclusive (doubtful), and greater than 50% was considered negative. In addition to the manufacturer’s positive and negative controls, internal control sera were also used as tracers according to the quality assurance system of the laboratory.

#### 2.5.2. Virus Neutralisation Test (VNT)

Positive and doubtful samples from c-ELISA were also screened for WNV- and USUV-neutralising antibodies using the VNT as described by Di Gennaro et al. [33]. This technique, which is based on the capability of the test serum to neutralise the cytopathic effect of the virus, is more specific than c-ELISA and produces fewer false positive results. Apart from detecting specific neutralising antibodies, this technique can also determine neutralising titres.

This technique was performed in cell culture microplates, using four wells per serum dilution. After inactivation for one hour at 56 °C, two 50 μL serum dilutions (from 1:5 to 1.640) were mixed with equal volumes of the virus, which contained 100 tissue culture infectious doses of 50% (TCID50). The plates were then incubated at 37 °C with 5% CO_2_ for 1 h. Positive and negative control sera were included in each plate. Vero cells grown in Dulbecco’s Modified Eagle’s Medium supplemented with 5% foetal calf serum were added to obtain confluence within 48 h.

To test virus activity, four replicates of the virus in TCID_50_ at concentrations of 1, 10, 100, and 1000 were included in each VNT. Readings were made on the fifth day by observing the presence and extension of the cytopathic effects (CPEs) in each well. Sera with a neutralising titre equal to or greater than 1:10 were considered positive.

### 2.6. Statistical Analysis for Horses’ Samples

Statistical analysis was carried out on horses’ samples. The collected data from horses’ samples were entered into a Microsoft Excel spreadsheet and analyzed using SPSS software. Descriptive analysis was carried out on all collected samples. SPSS statistical analysis (chi-square and logistic regression) was also utilized. Whereas for dogs’ samples, given the small sample size, only descriptive statistics were performed including counts and percentages.

Categorical variables including age group, sex, breed, and city were considered. The prevalence was calculated by dividing the number of ELISA positive samples by the total number of samples tested.

The seroprevalence for each city, sex, breed, and age group was calculated by dividing the number of ELISA positive samples from the total number of samples tested from that risk factor group. Univariable analysis using chi-square tests was conducted to measure the statistical significance of the association between the hypothesized risk factors (predictor variable) and the outcome variable (horse, diagnosed WNV). A forward conditional method, with sequential manual removal of variables based on a lack of statistical significance and biological plausibility, was used to produce the best-fitting model. For the factors to remain in the final model, the significance level was set at *p*-value = 0.05 for inclusion and 0.10 for exclusion.

## 3. Results

### 3.1. Seroprevalence of WNV in Horses

Out of 574 horse serum samples tested by c-ELISA, 76 (13.2%) were found to be positive for WNV antibodies and 4 were doubtful (Table 2). To confirm whether the reactivity to c-ELISA was due to the presence of WNV antibodies, the reactive samples (n = 80: 76 positive samples and 4 doubtful samples) underwent VNT. Specific WNV-neutralising antibodies were detected in 62 of the 80 tested serum samples (77.5%), representing 10.8% (n = 62/574) of the total tested horse samples, with titres ranging from 1:10 to 1:640 (Table 2). Although some samples showed high WNV neutralising titres, USUV neutralising antibodies were not detected in any of them.

### 3.2. Seroprevalence of WNV in Dogs

Out of the 63 dog serum samples tested by c-ELISA for the presence of WNV antibodies, 19 were found to be positive (Table 2). All of these positive samples underwent VNT. Specific WNV-neutralising antibodies were detected in 17 serum samples, representing 27% (17/63) of the total tested dog samples, with titres ranging from 1:10 to 1:320 (Table 2). No USUV neutralising antibodies were detected in the tested samples.

### 3.3. Risk Factor Analysis (ELISA IgG for Horses)

Concerning WNV seroprevalence related to horse breeds, Arabian horses showed the highest percentage of IgG seropositivity (20%; 29/145), followed by thoroughbred horses (14.5%; 43/295), and local Libyan horses (11.4%; 4/35) (Table 3). However, these differences were not significant (*p* = 0.341).

When the horses were grouped by geographic area, the area with the highest percentage of positive animals was Al-Zawia (n = 11/26; 42.30%), followed by Al-Swani (20.5%; n = 14/68), Gasr Ben Ghashir (15%; n = 21/140), Tripoli (14%; n = 22/157), Surman (10.7%; n = 3/28), Zawarah (8.9%; n = 5/56), and Al-Marj (0%; n = 0/99) (Table 4). Significant difference in WNV seroprevalence was found between western and eastern Libya.

For the purpose of this study, the horses were organised into five groups based on age range, as follows: younger than 6 months (n = 16), 7 to 18 months (n = 122), 19 to 48 months (n = 355), 49 to 72 months (n = 50), and older than 72 months (n = 31). WNV seropositivity was observed to increase with age (*p* = 0.0001) (Table 5).

After conducting the univariable analysis using chi-square tests to measure the statistical significance of association between the hypothesized risk factors (predictor variable) and the outcome variable (horse, diagnosed WNV). A forward conditional method applied on the significant risk factors, with sequential manual removal of variables based on a lack of statistical significance and biological plausibility, was used to produce the best-fitting model. For the factors to remain in the final model, the significance level was set at *p*-value = 0.05 for inclusion and 0.10 for exclusion. The significant risk factors (city, sex, breed, and age group) were included in the subsequent multivariable logistic regression model for horse samples. A forward stepwise (likelihood ratio) method with sequential manual removal of variables based on a lack of statistical significance and biological plausibility was used to produce the best-fitting model. For the factors to remain in the final model, the significance level was set at *p*-value = 0.05 for inclusion and 0.10 for exclusion. Age group was the only significant risk factor for horses in this study (Table 6).

## 4. Discussion

WNV has re-emerged globally as an important pathogen that affects humans and horses, with a distinct epidemiology and an irregular disease scenario [34,35,36]. Although recent global surveillance data have shown that the WNV-related incidence of neurological disease has increased and expanded geographically, with recurrent horse and human epidemics in many regions, the data available from Africa are still scant. There may be many reasons behind this lack of data, although we assume that a lack of funding is the most likely. However, even though the real burden of WNV infections in Africa is not well known, the little information currently available is sufficient to provide evidence that WNV originated on and is circulating across the continent [37].

Our results confirm that the virus has circulated or is circulating in Libya. WNV antibodies were detected in both horses and dogs. Only a few African countries, most of them sub-Saharan, have investigated the presence of WNV antibodies in dogs. A survey of dogs in South Africa revealed that 46% were positive for haemagglutination-inhibiting antibodies against WNV [38]. In Morocco, a study of military working dogs and horses reported similar seroprevalence in dogs (62%) and horses (60%), indicating that both species can be efficiently used as sentinel animals [25]. Interestingly, this study found that the prevalence in dogs (27%) was significantly (*p* < 0.05) higher than that in horses (10.8%). All dogs tested in this study were from Tripoli, while the horses originated from several areas in Libya. Based on their lifestyle, dogs can be regarded as good indicators of the circulation of WNV in urban areas, while horses, which normally live in the countryside, are good indicators of the circulation in rural areas. In our survey, it appears that the prevalence of WNV in the urban area was significantly higher than that in the rural area.

Different prevalence levels were also found between the western and eastern regions of Libya, with a higher prevalence in the former. One possible explanation for this is that the majority of samples collected from the eastern region were from younger horses. These findings show that the circulation of WNV in Libya is not uniformly distributed.

Apart from providing figures for WNV circulation in urban areas, monitoring dog populations that live near human populations could provide valuable information about the level of human exposure to WNV. Based on our findings, it seems that the population in Tripoli has been highly exposed to WNV infection. This situation is rather worrying even if, unlike dogs, people are likely to protect themselves from mosquito bites. A seroprevalence study of the presence of WNV IgG antibodies in 400 people in Tripoli using ELISA in 2017 found a prevalence of 2.75% (11/400), much lower than that detected in this study [30]. Therefore, even though blood samples from people in contact with the sampled animals were not taken, because that was beyond the scope of the current study, our results indicate that WNV is circulating in urban areas, and this should raise concern about a possible increase in human cases.

Despite the relatively high prevalence, clinical WND cases have not been reported in sampled animal populations or in humans [21,23]. As observed by other authors in many African countries, particularly the North African countries surrounding the Mediterranean basin, WND seems to be endemic, causing only mild, self-limited febrile disease [26,34,39,40,41,42].

The apparent absence of severe WND cases in Libya could also be a consequence of the circulation of relatively mild strains of WNV. The only way to confirm this hypothesis is to uncover the genome characteristics of the circulating WNV strain. This can be achieved only through thorough epidemiological investigation focused on humans and the comprehensive monitoring of vectors and reservoir hosts [31]. We thus strongly recommend that more research be carried out on WND in humans in Libya, focusing on the areas where there has been evidence of WNV circulating among animals. Evidence from Europe suggests that accurately identifying mosquito species in an area is important to reveal and predict the emergence of WNV in urban and rural environments. This is true for any surveillance strategy, including those for other zoonotic arboviruses [43].

Our survey demonstrated that dogs and horses in Libya have been exposed to WNV and possibly other closely related Flaviviruses. The VNT, which is the gold-standard method for WNV serology, can identify WNV-specific and cross-reacting antibodies. In our case, the negative results of the VNT against USUV excluded possible false positive results due to the co-circulation of this virus, which is cross-related to WNV. Out of the tested sera, there were eighteen horses’ sera and two dogs’ sera tested positive for WNV in the cELISA but were not confirmed in the neutralization test. Theoretically, these results could be explained by the presence of antibodies to the epitope of the E proteins of one of the Flaviviruses at an early stage of infection. They compete for the binding site of the detection antibody in the kit but are non-neutralizing and therefore are not detected in the VNT. Another potential explanation could be the existing cross-reactivity with other Flaviviruses unknown or not investigated in this study; however, no other studies on Flaviviruses have been conducted in Libya. 

The epidemiology of WNV and USUV has undergone dramatic changes over recent decades, with increases in the numbers of cases and the occurrence of outbreaks in different European countries [2]. In Europe, USUV has been serologically detected in horses and dogs, and USUV RNA has been detected in animal species such as bats, squirrels, wild boar, deer, and lizards [44]. Zoonotic concern about USUV causing neuroinvasive disease in humans in different countries has been reported with increasing frequency [45]. In this study, all samples were seronegative for USUV antibodies, implying that USUV has not circulated or is not circulating in Libya despite the suitable environmental conditions. We have also seen how variables such as the geographic area and the type of settlement have influenced WNV prevalence.

In this study, another factor that was found to increase the risk of being exposed to WNV infection was age. To validate our results, and also to remove the effect of ELISA false positives from the results, we generated another logistic regression using VNT as the dependent variable with the same risk factors and found that age group was the only risk factor that had a stronger association with positive ELISA. In addition, positive ELISA in the young age group (younger than 18 months) was 100% negative by VNT (Table 5). Even sex was significant (*p* = 0.021), but it was excluded as a significant risk factor for biological reasons; in addition, the average age of males was younger than the average age of females (data not shown). This finding has been widely reported in horses. A study from Egypt revealed that horses aged ≥15 years, stallions, and mixed-breed horses had potential risk factors associated with high WNV seroprevalence rates [9]. Another survey showed that age, together with other variables such as the presence of ponds, the use of insecticides and the presence of rice fields and ruminants on the same property, increased the exposure of horses and wild birds to WNV [46].

Other factors were frequently identified as being associated with WNV seroprevalence in horses, including low numbers of horses within holdings, the travel patterns and presence of mosquitoes [47], the presence of dead birds and sick animals on a property, the use of fans, and the use of stables constructed of solid wood or cement [48].

## 5. Conclusions

The present study provides new evidence about the circulation of WNV in animal populations in certain environments of this country. This adds to the knowledge of the ongoing documented endemic status of the virus in North Africa and its possible emergence as an important human health problem.

There is an urgent need for continuous monitoring programmes for humans, horses, mosquitoes, and birds, including migrating avian species, in order to provide essential epidemiological data for the early detection of WNV and its circulation patterns. It is equally crucial to increase public and professional awareness about WNV and associated clinical problems in animals and humans.

## Figures and Tables

**Figure 1 pathogens-13-00041-f001:**
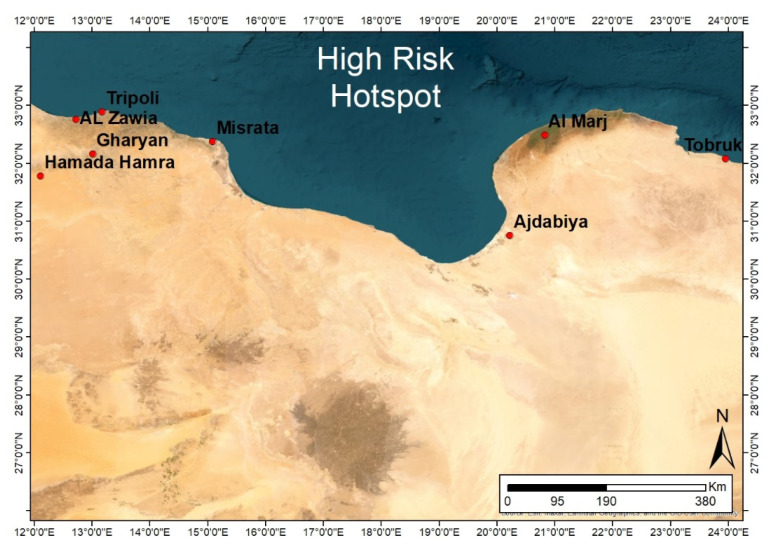
Map of Libya indicating the eight hotspot locations (Tobruk, Al Marj, Ajdabiya, Misrata, Tripoli, Al Zawia, Gharyan, and Hamada Hamra). An ecological environment that is suitable for the life cycle of the virus and the presence of the vector.

**Figure 2 pathogens-13-00041-f002:**
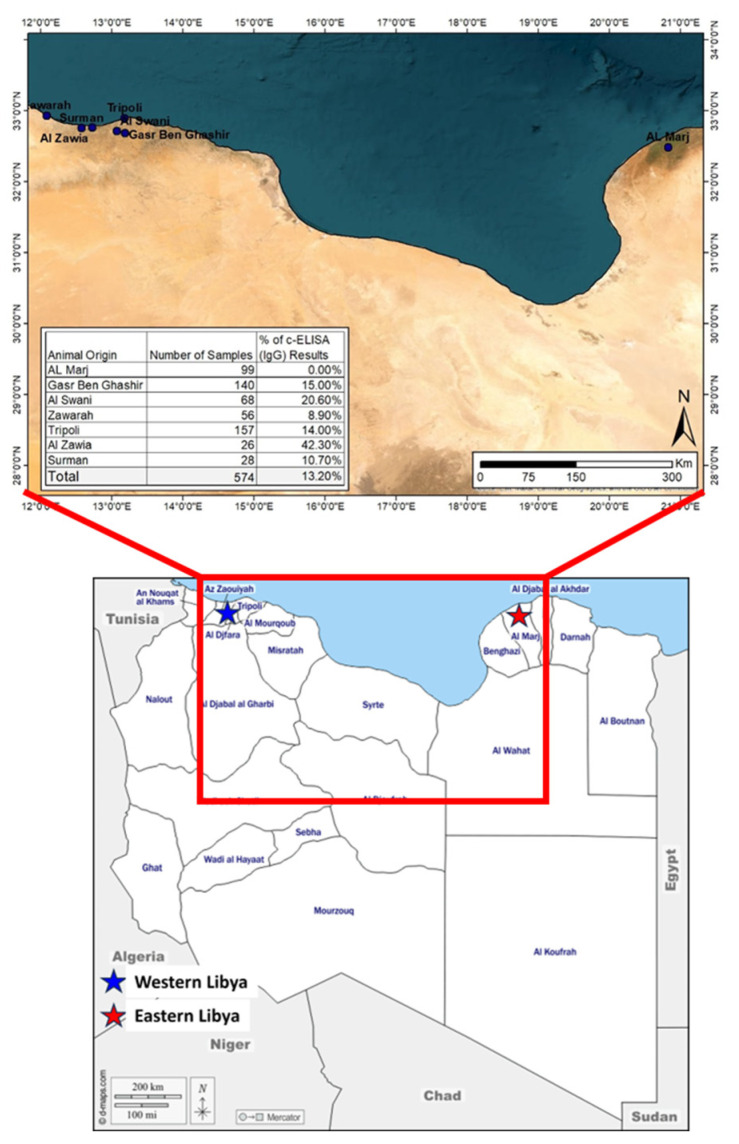
Map of Libya indicating sites of samples collection from eastern and western Libya. In Al Zawia area (Surman, Al Zawia, and Zawarah), in Tripoli area (Tripoli, Gasr Ben Ghashir, and Al Swani) in western Libya and in Al Marj in eastern Libya.

**Table 1 pathogens-13-00041-t001:** Demographic data of this study’s sample animals.

Total Number of Samples	Age (Months)	Sex	Animal Breed	Area
Horses (n = 574)	2–240	49.3% male (n = 283) 50.7% female (n = 291)	Arabian (n = 145, 25.3%)	Western Libya
			Local thoroughbred (n = 202, 35.2%)	
			Imported thoroughbred (n = 93, 16.2%)	
			Local Libyan (n = 35, 6.1%)	
			Mixed (n = 99, 17.2%)	Eastern Libya
Dogs (n = 63)	3–72	49.2% male (n = 31) 50.8% female (n = 32)	Many breeds	Tripoli

**Table 2 pathogens-13-00041-t002:** West Nile virus (WNV) seroprevalence in Libyan horses and dogs.

Total Samples	Positive Samples		VNT Titre Range
	c-ELISA	VNT	
Horses (n = 574)	13.2% (n = 76/574)	Of positive ELISA: n = 62/80 (77.5%) Of total samples: n = 62/574 (10.8%)	1:10–1:640
Dogs (n = 63)	30.2% (n = 19/63)	Of positive ELISA: n = 17/19 (89.5%) Of total samples: n = 17/63 (27%)	1:10–1:320

c-ELISA, competitive enzyme-linked immunosorbent assay; VNT, virus neutralisation test.

**Table 3 pathogens-13-00041-t003:** WNV seroprevalence in Libyan horses based on breed in western part of Libya.

Breed		ELISA IgG		Total
		Negative	Positive	
Arabian	Count	116	29	145
	% within breed	80%	20%	100.0%
	% of total	24.4%	6.1%	30.5%
Local Thoroughbred	Count	172	30	202
	% within breed	85.1%	14.9%	100.0%
	% of total	36.2%	6.3%	42.5%
Imported Thoroughbred	Count	80	13	93
	% within breed	86.0%	14.0%	100.0%
	% of total	16.8%	2.7%	19.6%
Libyan	Count	31	4	35
	% within breed	88.6%	11.4%	100.0%
	% of total	6.5%	0.8%	7.4%
Total *	Count	399	76	475
	% within breed	84%	16%	100.0%
	% of total	84%	16%	100.0%

* Statistical analysis excluded samples from Al-Marj (n = 99) due to seronegative results.

**Table 4 pathogens-13-00041-t004:** WNV seroprevalence in Libyan horses according to geographic area.

Animal	Animal Origin (City)	c-ELISA (IgG) Results	
		Negative	Positive
Horses	Al-Marj ^a^ (n = 99)	n = 99 (100%)	n = 0 (0%)
	Gasr Ben Ghashir ^b^ (n = 140)	n = 119 (85%)	n = 21 (15%)
	Al-Swani ^b^ (n = 68)	n = 54 (79.4%)	n = 14 (20.6%)
	Zawarah ^b^ (n = 56)	n = 51 (91.1%)	n = 5 (8.9%)
	Tripoli ^b^ (n = 157)	n = 135 (86%)	n = 22 (14%)
	Al-Zawia ^b^ (n = 26)	n = 15 (57.7%)	n = 11 (42.3%)
	Surman ^b^ (n = 28)	n = 25 (89.3%)	n = 3 (10.7%)
	Total (574)	n = 498 (86.8%)	n = 76 (13.2%)
Dogs	Tripoli ^b^ (n = 63)	n = 44 (69.8%)	n = 19 (30.2%)

^a^ Located in eastern region of Libya; ^b^ located in western region of Libya.

**Table 5 pathogens-13-00041-t005:** WNV seroprevalence in Libyan horses according to age.

Age Group		ELISA IgG			VNT		Total
		Negative	Positive	Doubtful	Negative	Positive	
<6 months	Count	16	0	0	0	0	16
	% within age group	100.0%	0.0%	0.0%	N/D	N/D	100.0%
7–18 months	Count	115	5	2	7	0	122
	% within age group	94.2%	4.1%	1.6%	100%	0.0%	100.0%
19–48 months	Count	303	51	1	7	45	355
	% within age group	85.4%	14.4%	0.2%	13.5%	86.5%	100.0%
49–72 months	Count	39	10	1	1	10	50
	% within age group	78%	20%	2%	9.1%	90.9%	100.0%
>72 months	Count	21	10	0	3	7	31
	% within age group	67.7%	32.3%	0.0%	30%	70%	100.0%
Total	Count	494	76	4	18	62	574
	% within age group	86.1%	13.2%	0.7%	22.5%	77.5%	100.0%

**Table 6 pathogens-13-00041-t006:** Variables in final equation for horse model.

Variables in Equation							
		B	SE	Wald	df	*p*-Value	Exp(B)
Step 1 ^a^	Age group	0.698	0.150	21.610	1	0.0001	2.010
	Constant	−4.017	0.501	64.399	1	0.0001	0.018
Step 2 ^b^	Age group	0.661	0.157	17.760	1	0.0001	1.936
	Sex	0.610	0.265	5.298	1	0.021	1.841
	Constant	−4.874	0.655	55.443	1	0.0001	0.008

^a^ Variable entered in step 1: Age_group; ^b^ variable entered in step 2: Sex. *p* < 0.05: Statistically significant.

## Data Availability

All data supporting the findings of this study are available within this manuscript. Any additional data can be provided by the corresponding author upon reasonable request.

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
