# Peer review of "Evidence of West Nile Virus Circulation in Horses and Dogs in Libya"

_pathogens, 2023, doi:10.3390/pathogens13010041_

Round 1
Reviewer 1 Report (New Reviewer)
Comments and Suggestions for Authors
The manuscript and its conclusions provide a good contribution to the surveillance of the spread of West Nile virus (WNV) in North African countries. The obtained serologic ELISA results are supported by the VNT results. This is an important step in confirming the results.
The manuscript is easy to read, although a careful revision regarding formatting seems necessary (e.g. presentation of the equation in line 132).
An important point for revision is the statistical analysis that leads to the results of the assessment of the risk factors. The equation used and the significance of the variables in Table 6 should be shown here. Without this additional information, the data loses its value.
On the latter point, the manuscript still needs to be improved before it can be published.
Author Response
Reviewer 1
The manuscript and its conclusions provide a good contribution to the surveillance of the spread of West Nile virus (WNV) in North African countries. The obtained serologic ELISA results are supported by the VNT results. This is an important step in confirming the results.
The manuscript is easy to read, although a careful revision regarding formatting seems necessary (e.g. presentation of the equation in line 132).
Answer:
Presentation of the equation has been done in a proper way.
An important point for revision is the statistical analysis that leads to the results of the assessment of the risk factors. The equation used and the significance of the variables in Table 6 should be shown here. Without this additional information, the data loses its value.
Answer:
After conducting the univariable analysis using chi-square tests to measure the statistical significance of association between the hypothesized risk factors (predictor variable) and the outcome variable (horse, diagnosed WNV). A forward conditional method applied on the significant risk factors, with sequential manual removal of variables based on a lack of statistical significance and biological plausibility, was used to produce the best-fitting model. For the factors to remain in the final model, the significance level was set at p-value = 0.05 for inclusion and 0.10 for exclusion. The significant risk factors (city, sex, breed and age group) were included in the subsequent multivariable logistic regression model for horse samples. A forward stepwise (likelihood ratio) method with sequential manual removal of variables based on a lack of statistical significance and biological plausibility was used to produce the best-fitting model. For the factors to remain in the final model, the significance level was set at p-value = 0.05 for inclusion and 0.10 for exclusion. Age group was the only significant risk factor for horses in this study (Table 6).
On the latter point, the manuscript still needs to be improved before it can be published.
Answer:
We have responded to all three reviewers’ comments and correction in order to improve our manuscript. Hoping that all we have done will satisfy all respected reviewers.

Reviewer 2 Report (New Reviewer)
Comments and Suggestions for Authors
Dear authors,
Thank you for the interesting article on identifying serological markers for the presence of the West Nile virus in Libya.
In your work, you were able to obtain convincing evidence of the presence of the West Nile virus in Libya. An important part of the evidence is the information you receive about the presence of virus-neutralizing antibodies detected by a neutralization test.
However, in my opinion, the article requires revision.
Main comments
The analysis of risk factors yielded only a connection with the age of the animals. This is a fairly expected result: the older the animal, the greater its exposure to mosquito bites, and as a result, the greater the likelihood of contact with mosquito pathogens. The remaining risk factors selected were negative, and the biological meaning of such a negative result is highly questionable. I propose to remove this section on risk factor analysis from the article.
Instead of risk factors, I recommend expanding the analysis of the results obtained by adding detailed information on the distribution of seropositive horses in different provinces (Al Marj, Al Zawia, and Tripoli areas). It could be interesting to divide the existing sample according to the criteria: city - village/rural area.
Small comments
Line 28. Please replace “Usutu virus antibodies” with “other flavivirus antibodies”. There is a possibility that other flaviviruses may be circulating in Libya and will give a cross-reaction in the ELISA.
Lines 28-30. First you write that you found 76/574 IgG-positive samples, but then you add that in the neutralization test, you examined 80 samples (instead of 76). It's confusing. You explain this in more detail in the Methods section, but many people read only the Abstract, and therefore this sentence may cause misunderstanding. Consider rewriting this part of the Abstract.
Moreover, I propose not to indicate the percentage of ELISA-positive results in the Abstract. Instead, I would only report the percentage of samples confirmed to be neutralized (out of total samples) - 10.8% for horses and 26.9% for dogs.
Line 64. Please cite Guthrie et al., 2003 according to journal requirements.
Line 77. You write about clinical manifestations in patients and cite an article from Bertram et al 2020. However, in this article, the authors study horses, not patients. Therefore, you should either replace the reference to the relevant article or revise the sentence.
Line 103. Many readers of the article are poorly familiar with the geography of Libya, and it is difficult to visualize the location of the places, cities, and provinces mentioned in the Study design section. It will be very useful if the authors add a map of Libya to this section, indicating the eight hotspots, sample collection sites, as well as geographic attributes relevant to WNV ecology:
• presence of reservoirs, rivers/arid areas;
• population density in different regions.
Line 110. I think there is a typo here, Al Marj is located in eastern Libya, not western.
Line 118. Please explain what 95% means.
Lines 126-133. Please provide a link to the source according to which you did the sample calculation, and where I can find out the details of this approach to sample calculation.
Line 176. I think it will be useful if you list in the text which flaviviruses this ELISA kit detects (according to the manufacturer’s instructions), and also please add an explanation that ... designed to detect multispecies antibodies (including horses and dogs).
Line 238. As I wrote above, I recommend revising this block of work and either choosing other risk factors or completely eliminating this section from the work.
Author Response
Reviewer 2
Dear authors,
Thank you for the interesting article on identifying serological markers for the presence of the West Nile virus in Libya.
In your work, you were able to obtain convincing evidence of the presence of the West Nile virus in Libya. An important part of the evidence is the information you receive about the presence of virus-neutralizing antibodies detected by a neutralization test.
However, in my opinion, the article requires revision.
Main comments
The analysis of risk factors yielded only a connection with the age of the animals. This is a fairly expected result: the older the animal, the greater its exposure to mosquito bites, and as a result, the greater the likelihood of contact with mosquito pathogens. The remaining risk factors selected were negative, and the biological meaning of such a negative result is highly questionable. I propose to remove this section on risk factor analysis from the article.
Answer:
Thank you for your valuable comment. However, the other two reviewers insisted not only to keep the risk factors but also to be added to the Abstract. In addition to add univariable analysis in the paper.
Instead of risk factors, I recommend expanding the analysis of the results obtained by adding detailed information on the distribution of seropositive horses in different provinces (Al Marj, Al Zawia, and Tripoli areas). It could be interesting to divide the existing sample according to the criteria: city - village/rural area.
Answer:
We have used the proportional sampling method in order to estimate the number of samples per area taking in the account the number of total horse population in each area. This was calculated from each selected area based on the number of horses in the area. Also, as recommended, we have added a map of Libya indicating all eight hotspots.
Small comments
Line 28. Please replace “Usutu virus antibodies” with “other flavivirus antibodies”. There is a possibility that other flaviviruses may be circulating in Libya and will give a cross-reaction in the ELISA.
Answer:
Thank you for your comment and the sentence has been corrected as suggested by you “Positive and doubtful sera were also tested using a more specific virus neutralisation assay to confirm whether the ELISA-positive results were due to WNV or other flavivirus antibodies.”
Lines 28-30. First you write that you found 76/574 IgG-positive samples, but then you add that in the neutralization test, you examined 80 samples (instead of 76). It's confusing. You explain this in more detail in the Methods section, but many people read only the Abstract, and therefore this sentence may cause misunderstanding. Consider rewriting this part of the Abstract. Moreover, I propose not to indicate the percentage of ELISA-positive results in the Abstract. Instead, I would only report the percentage of samples confirmed to be neutralized (out of total samples) - 10.8% for horses and 26.9% for dogs.
Answer:
Agree. The sentence has been rewritten as follows “The seroprevalence of WNV IgG antibodies according to ELISA was 13.2% out of 574 of total horses’ samples and 30.2% out of 63 of total dogs’ samples. The virus neutralisation test (VNT) confirmed that 10.8% (62/574) and 26.9% (17/63) were positive for WNV-neutralising titres ranging from 1:10 to 1:640.”
Line 64. Please cite Guthrie et al., 2003 according to journal requirements.
Answer:
The reference has been corrected and has been added to the References list as well.
Line 77. You write about clinical manifestations in patients and cite an article from Bertram et al 2020. However, in this article, the authors study horses, not patients. Therefore, you should either replace the reference to the relevant article or revise the sentence.
Answer:
You’re right. The sentence has been revised as follows “In a study in South Africa, 52% of horses positive for WNV had fever, 92% displayed neurological signs and 39% died [16].”
Line 103. Many readers of the article are poorly familiar with the geography of Libya, and it is difficult to visualize the location of the places, cities, and provinces mentioned in the Study design section. It will be very useful if the authors add a map of Libya to this section, indicating the eight hotspots, sample collection sites, as well as geographic attributes relevant to WNV ecology:
- presence of reservoirs, rivers/arid areas;
- population density in different regions.
Answer:
Well noted. A map of Libya has been added in Figure as suggested by the reviewer.

Figure 1. Map of Libya indicating the eight hotspot locations; 1: Tobruk; 2: Al Marj; 3: Ajdabiya; 4: Misrata; 5: Tripoli; 6: Al Zawai; 7: Gharyan; 8: Hamada Al Hamra districts. As ecological environment that is suitable for the life cycle of the virus and the presence of the vector.
Line 110. I think there is a typo here, Al Marj is located in eastern Libya, not western.
Answer:
You’re right. It has been corrected “Al Marj in eastern Libya”
Line 118. Please explain what 95% means.
Answer:
If we repeat taking random samples in each area with the same number of samples, 95% of the estimation of seroprevalence would be within this interval level.
Lines 126-133. Please provide a link to the source according to which you did the sample calculation, and where I can find out the details of this approach to sample calculation.
Answer:
The source for the sample calculation is a mobile application called “Statistics and Sample Size”. The link: https://play.google.com/store/apps/details?id=thaithanhtruc.info.sass&hl=en&gl=US
Line 176. I think it will be useful if you list in the text which flaviviruses this ELISA kit detects (according to the manufacturer’s instructions), and also please add an explanation that ... designed to detect multispecies antibodies (including horses and dogs).
Answer:
Agree. The paragraph has been rephrased as follows “The test is directed against an epitope of the E protein that is common to WNV and other Flaviviruses and proven to detect a wide range of Flavivirus antibodies including WNV, JEV, tick-borne encephalitis virus (TBEV), USUV, Zika virus (ZIKAV), and Dengue virus (DENV) in multiple species, including humans, horses, dogs, birds and others.”
Line 238. As I wrote above, I recommend revising this block of work and either choosing other risk factors or completely eliminating this section from the work.
Answer:
Thank you for your valuable comment. However, the other two reviewers insisted not only to keep the risk factors but also to be added to the Abstract. In addition to add univariable analysis in the paper.

Reviewer 3 Report (New Reviewer)
Comments and Suggestions for Authors
The work written by Ben-Mostafa et al. and titled "Evidence of West Nile virus circulation in horses and dogs in Libya" describes the presence of antibodies against WNV in the equine and canine population in Libya for the first time. The approach used involved the use of a commercial ELISA and confirmation by virus neutralization. The work is original, deals with a current topic of an emerging infection and is written with scientific rigor. Some points need to be clarified (listed below).
Line 23: How did the authors certify that the animals were healthy? Maybe better apparently healthy?
Line 32: The authors speak of a "subsequent multivariate analysis" without mentioning the chi-square analysis previously in the abstract.
Line 33: Unnecessary p value, please delete.
Line 53: Species in which seroconversion has been found could be listed. Below are some studies that the authors might be helpful in listing these species. doi: 10.3390/ani10030494; doi: 10.1007/s11250-020-02339-x; doi: 10.1007/s00705-020-04719-y; doi: 10.1007/s00705-004-0478-5
Line 96-100: Rephrase and put before the aim of the work.
Line 131: Reference for sampling type and formula.
In general: several parts in yellow and text formatting problems (size and font). Please standardize.
Table 1: Unnecessary table. You can move it to supll files.
Line 231: “(27%; 95% CI: 17.6–39.1%)” delete.
Line 236: Explain why risk analysis has not been carried out for the canine species (sex, age, etc.).
Comparison with other prevalences obtained in the world (Corsica region of France, Mexico, Campani region of Italy, Vietnam, Argentina, Spain)
Comments on the Quality of English LanguageThe manuscript is written clearly and with a good level of English.
Author Response
Reviewer 3
The work written by Ben-Mostafa et al. and titled "Evidence of West Nile virus circulation in horses and dogs in Libya" describes the presence of antibodies against WNV in the equine and canine population in Libya for the first time. The approach used involved the use of a commercial ELISA and confirmation by virus neutralization. The work is original, deals with a current topic of an emerging infection and is written with scientific rigor. Some points need to be clarified (listed below).
Line 23: How did the authors certify that the animals were healthy? Maybe better apparently healthy?
Answer:
Agree. The sentence has been corrected “from apparently healthy,”
Line 32: The authors speak of a "subsequent multivariate analysis" without mentioning the chi-square analysis previously in the abstract.
Answer:
Well noted. We have added that to the Abstract as follows “Univariable analysis using chi-square tests was conducted to measure the statistical significance of association between the hypothesized risk factors including city, sex, breed and age group then were analyzed using the subsequent multivariable logistic regression model for horse samples.”
Line 33: Unnecessary p value, please delete.
Answer:
P value has been deleted.
Line 53: Species in which seroconversion has been found could be listed. Below are some studies that the authors might be helpful in listing these species. doi: 10.3390/ani10030494; doi: 10.1007/s11250-020-02339-x; doi: 10.1007/s00705-020-04719-y; doi: 10.1007/s00705-004-0478-5
Answer:
Well noted. Suggested References have been added.
Line 96-100: Rephrase and put before the aim of the work.
Answer:
Well noted. The paragraph has been rephrased as suggested by the reviewer.
Line 131: Reference for sampling type and formula.
Answer:
Sample type was proportional sampling. The Reference is the book: “Veterinary epidemiology: principles and methods” by: S. Wayne Martin, Alan H Meek, Preben Willeberg.
The source for the sample calculation is a mobile application called “Statistics and Sample Size”. The link: https://play.google.com/store/apps/details?id=thaithanhtruc.info.sass&hl=en&gl=US
In general: several parts in yellow and text formatting problems (size and font). Please standardize.
Answer:
Done and all font and size have been standardized.
Table 1: Unnecessary table. You can move it to supll files.
Answer:
Well noted. I’ll leave to journal’s editor to move it as Supplementary or to keep it.
Line 231: “(27%; 95% CI: 17.6–39.1%)” delete.
Answer:
Deleted.
Line 236: Explain why risk analysis has not been carried out for the canine species (sex, age, etc.).
Answer:
Because the sample number of dogs (63) was too small with 5 breeds of dogs. However, we have done already the univariable analysis and found there was no significant for age, breed and sex. The location is not important as all from Tripoli.
Comparison with other prevalences obtained in the world (Corsica region of France, Mexico, Campani region of Italy, Vietnam, Argentina, Spain)
Answer:
Done. New References have been added.

Round 2
Reviewer 2 Report (New Reviewer)
Comments and Suggestions for Authors
I appreciate the efforts of the authors in revising their manuscript, and I think this work has been beneficial. However, there are still minor flaws that require additional revision.
1. The added map makes it very easy to perceive the results obtained. However, there are several comments on the map:
The map shows not only administrative and state boundaries, but also additional information, the meaning of which is unclear and is not disclosed either in the legend or in the caption to the figure. The map clearly shows areas marked in different colors (from blue to brown). If this information is relevant to the study, carries auxiliary information (population density, or relief, or amount of precipitation), then this should be deciphered in the legend. If this information is not relevant to the research, then it is better to remove these colors from the map.
The same goes for red dots.
Additionally, the hotspots mentioned in the study and sample collection sites are located along the coast in the north of the country. At this scale, these hot spots are tightly concentrated, making it difficult to perceive. I recommend that the authors add another inset map that shows only the northern coast in a larger view. This will make it easier to understand.
It is difficult to estimate the scale from this map, for example, to find out the distance between the western and eastern regions of the country. Please add a scale bar to the map.
You collected samples in 3 locations. I think it will be useful to indicate them on the map with a separate symbol.
Line 127. Again, I have a question about this sentence. The Guesstimate approach is relatively rarely used in the scientific community. And as a result, this approach will be new to many readers. Someone might want to replicate your approach to sampling design. Therefore, it is important to describe the methods used so that other teams can reproduce them. Please include reference in your manuscript that describes the guesstimate approach.
As far as I understand, this is an expert assessment of prevalence, based on available literature data. However, I cannot understand how statistical measures of confidence can be added to such "guess-data". You answered: “If we repeat taking random samples in each area with the same number of samples, 95% of the estimation of seroprevalence would be within this interval level.” That is, in essence, you are using bootstrapping for some kind of procedure for assessing seroprevalence. Please detail in the Methods section from which population/data you performed the “repeat taking random samples”, and also indicate the bootstrap parameters (number of iterations, software in which the calculations were carried out).
Table 2. Last line: I calculated 17/63 and got 27.0 (instead of 26.9) - this, of course, is not a big difference, but science loves accuracy.
Line 382. Please remove the word "vector". In this work you did not study arthropod vectors.
Author Response
Reviewer 2
I appreciate the efforts of the authors in revising their manuscript, and I think this work has been beneficial. However, there are still minor flaws that require additional revision.
The added map makes it very easy to perceive the results obtained. However, there are several comments on the map:
The map shows not only administrative and state boundaries, but also additional information, the meaning of which is unclear and is not disclosed either in the legend or in the caption to the figure. The map clearly shows areas marked in different colors (from blue to brown). If this information is relevant to the study, carries auxiliary information (population density, or relief, or amount of precipitation), then this should be deciphered in the legend. If this information is not relevant to the research, then it is better to remove these colors from the map. The same goes for red dots.
Additionally, the hotspots mentioned in the study and sample collection sites are located along the coast in the north of the country. At this scale, these hot spots are tightly concentrated, making it difficult to perceive. I recommend that the authors add another inset map that shows only the northern coast in a larger view. This will make it easier to understand.
It is difficult to estimate the scale from this map, for example, to find out the distance between the western and eastern regions of the country. Please add a scale bar to the map.
You collected samples in 3 locations. I think it will be useful to indicate them on the map with a separate symbol.
Answer:
Thank you for your valuable advice. We have created two new maps of Libya indicating the hotspots and samples collection sites and removed all unnecessary information.

Figure 1. Map of Libya indicating the eight hotspot locations (Tobruk, Al Marj, Ajdabiya, Misrata, Tripoli, Al Zawia, Gharyan and Hamada Hamra). As ecological environment that is suitable for the life cycle of the virus and the presence of the vector.

Figure 2. Map of Libya indicating sites of samples collection from Al Marj in eastern Libya, and from Al Zawia area (Surman, Al Zawia and Zawarah) and from Tripoli area (Tripoli, Gasr Ben Ghashir and Al Swani) in western Libya.
Line 127. Again, I have a question about this sentence. The Guesstimate approach is relatively rarely used in the scientific community. And as a result, this approach will be new to many readers. Someone might want to replicate your approach to sampling design. Therefore, it is important to describe the methods used so that other teams can reproduce them. Please include reference in your manuscript that describes the guesstimate approach.
As far as I understand, this is an expert assessment of prevalence, based on available literature data. However, I cannot understand how statistical measures of confidence can be added to such "guess-data". You answered: “If we repeat taking random samples in each area with the same number of samples, 95% of the estimation of seroprevalence would be within this interval level.” That is, in essence, you are using bootstrapping for some kind of procedure for assessing seroprevalence. Please detail in the Methods section from which population/data you performed the “repeat taking random samples”, and also indicate the bootstrap parameters (number of iterations, software in which the calculations were carried out).
Answer:
Well noted and you are right. We have changed the “guesstimate” approach to “proximate” approach. Sampling type was proportional sampling. The Reference is the following book:
Martin, S.W.; Meek, A.H. Veterinary Epidemiology: Principles and Methods. In: Sampling Methods. Preben Willeberg. Iowa State University Press, USA,1987, pp: 22-47
The above Reference has been added to the References list as suggested by you.
Table 2. Last line: I calculated 17/63 and got 27.0 (instead of 26.9) - this, of course, is not a big difference, but science loves accuracy.
Answer:
Agree. The percentage has been corrected to 27%.
Line 382. Please remove the word "vector". In this work you did not study arthropod vectors.
Answer:
Agree. The word “vector” has been removed.

This manuscript is a resubmission of an earlier submission. The following is a list of the peer review reports and author responses from that submission.
Round 1
Reviewer 1 Report
Comments and Suggestions for Authors
The study on "Evidence of West Nile virus circulation in horses and dogs in Libya" carries vital significance as it offers crucial insights into the potential reservoirs and vectors of this virus, informing public health efforts to monitor and control its spread. Understanding the presence of West Nile virus in horses and dogs not only aids in assessing the risk to these animals but also raises awareness about the zoonotic potential, thereby enhancing disease surveillance, prevention, and management strategies. This research holds the potential to safeguard both animal and human health by guiding targeted interventions and reinforcing the importance of “One Health” approaches in infectious disease control.
Concerning the article's quality, there exist a multitude of typographical and grammatical inaccuracies, along with technical writing problems within the manuscript. I would recommend that the author thoroughly review the manuscript to address these issues and consider possible revisions.
Some of the minor/major concerns are as follows:
Abstract: Add statistical values, where applicable.
Keywords: Rephrase as “West Nile Virus; Horses; Dogs; Seroprevalence; Libya”.
Introduction:
Include details regarding the pathogenesis and the significance (clinical consequences) of West Nile Virus infection in horses and dogs.
Line No. 62: “Not mammal species” Is this phrasing accurate?
Line No. 70-71: Rephrase the sentence.
Materials and Methods:
Line No. 84: Was there a specific reason for the extended three-year duration needed to complete this seroprevalence study?
> Why did the authors opt for collecting 574 blood samples from horses while only collecting 63 from dogs? As an epidemiological study, did the authors employ any particular epidemiological formula for determining the sample sizes?
> Likewise, what criteria were employed for selecting samples within breeds and across various regions?
> Since Table 2 encompasses the results of Tables 3 and 4, I recommend excluding Tables 3 and 4 from the article.
> Why variables such as breed and age were presented for horses only but not for dogs? Could you please provide an explanation for this discrepancy?
> Upon reviewing the Results and Discussion sections, it appears that the results were not presented based on statistical analysis. Merely discussing the results based on percentage values, without taking into account the statistical analysis, lacks scientific rigor and will not provide a meaningful contribution to the scientific community.
Discussion:
> Line No. 230-233 and 233-237: Rephrase the sentences.
> Line No. 252-254: Add reference.
> In discussion section, also add explanation regarding the significant findings of the study, such as the seroprevalence in the western and eastern regions of Libya, as well as the seroprevalence observed in the higher age groups.
Conclusion:
Conclusion is poorly written. Please provide a concise conclusion based on the study's significant findings and avoid unnecessary explanation.
Comments on the Quality of English LanguageExtensive English editing required.
Author Response
Reviewer 1:
Thank you so very much for your feedback and excellent report in order to improve our manuscript. All comments have been taken in our revised manuscript and the manuscript has been sent to the MDPI Language Editor as well.
Please bear in mind that the study and samples collection have been carried out in Libya; the country that is unstable with conflict and war! We faced difficult times during our work, but with our goal to do great research we continued to go forward to our goal!
The study on "Evidence of West Nile virus circulation in horses and dogs in Libya" carries vital significance as it offers crucial insights into the potential reservoirs and vectors of this virus, informing public health efforts to monitor and control its spread. Understanding the presence of West Nile virus in horses and dogs not only aids in assessing the risk to these animals but also raises awareness about the zoonotic potential, thereby enhancing disease surveillance, prevention, and management strategies. This research holds the potential to safeguard both animal and human health by guiding targeted interventions and reinforcing the importance of “One Health” approaches in infectious disease control.
Concerning the article's quality, there exist a multitude of typographical and grammatical inaccuracies, along with technical writing problems within the manuscript. I would recommend that the author thoroughly review the manuscript to address these issues and consider possible revisions.
Some of the minor/major concerns are as follows:
Abstract: Add statistical values, where applicable.
Answer: Statistical values have been added in the main article.
Keywords: Rephrase as “West Nile Virus; Horses; Dogs; Seroprevalence; Libya”.
Answer: Done!
Introduction:
Include details regarding the pathogenesis and the significance (clinical consequences) of West Nile Virus infection in horses and dogs.
Answer: Provided!
Line No. 62: “Not mammal species” Is this phrasing accurate?
Answer: In other mammal and nonmammal species including dogs, cattle, sheep, goats, camels, deer, squirrels and reptiles, WNV infection can elicit antibodies.
Line No. 70-71: Rephrase the sentence.
Answer: Done!
Materials and Methods:
Line No. 84: Was there a specific reason for the extended three-year duration needed to complete this seroprevalence study?
Answer: Yes, the reason that we are working in unstable country “Libya” with conflict and war. It was not easy to go out and collect samples due to security reasons.
> Why did the authors opt for collecting 574 blood samples from horses while only collecting 63 from dogs? As an epidemiological study, did the authors employ any particular epidemiological formula for determining the sample sizes?
Answer: The formula for sample collection has been included in the manuscript. See section “2.2 Targeted animals and sampling strategy”.
> Likewise, what criteria were employed for selecting samples within breeds and across various regions?
Answer: The criteria has been included in the manuscript. See section “2.1 Study design”
> Since Table 2 encompasses the results of Tables 3 and 4, I recommend excluding Tables 3 and 4 from the article.
Answer: Accepted and Tables 3 and 4 have been removed!
> Why variables such as breed and age were presented for horses only but not for dogs? Could you please provide an explanation for this discrepancy?
Answer: For dogs, few samples were collected from Tripoli only, depending on the results of the single paper published on WND in humans in Libya, so dogs’ samples were collected from dogs living in the same area where the positive cases of human WND have been reported, assuming that dogs can be considered as good sentinels for monitoring WNV.
> Upon reviewing the Results and Discussion sections, it appears that the results were not presented based on statistical analysis. Merely discussing the results based on percentage values, without taking into account the statistical analysis, lacks scientific rigor and will not provide a meaningful contribution to the scientific community.
Answer: Statistical analysis has been added in more details in the Materials and Methods, Results and Discussion section in the manuscript!
Discussion:
> Line No. 230-233 and 233-237: Rephrase the sentences.
Answer: Done!
> Line No. 252-254: Add reference.
Answer: Done!
> In discussion section, also add explanation regarding the significant findings of the study, such as the seroprevalence in the western and eastern regions of Libya, as well as the seroprevalence observed in the higher age groups.
Answer: Done!
Conclusion:
Conclusion is poorly written. Please provide a concise conclusion based on the study's significant findings and avoid unnecessary explanation.
Answer: Done!
Reviewer 2 Report
Comments and Suggestions for Authors
In this manuscript, the authors analyses the prevalence of West Nile virus antibodies in horses and dogs in Libya. This sort of data is necessary to understand the circulation of the virus in North Africa and improve the public health preparation for potential West Nile virus outbreaks. On the positive side of the study it is very good that the authors confirm the detection of antibodies by seroneutralization because ELISA can generate a variable number of false positives due to cross reaction with antibodies generated by other virus that do not need to be of public health concern. However, the manuscript need important improvements in the introduction, methods, results and discussion of the results.
In the abstract, what is important is that this study allows to determine the levels of circulation of WNV in Libya because human cases of WNV are difficult to diagnose and use to be severely underreported. See also below my comments on discussing only the results of seroneutralization tests.
Avian species are key players in the transmission of the virus and their importance and involvement in the transmission cycle is not clearly stated in the introduction where only mammals and reptiles are mentioned.
In the methods section you need to clarify how you decided if the antibodies correspond to West Nile virus or to Usutu. You mention that only West Nile virus antibodies were found but it is highly unlikely that all the seroneutralizations were negative for Usutu so I presume that some of them were positive but the titres for West Nile were higher. Usually a threshold is used of four fold differences in titre to consider them specific for one or the other virus. I also suggest that you do not repeat the analyses using ELISA positive or Seroneutralization positives, this is very repetitive and confussing for the reader. Please, consider only positives by seroneutralization as positives and do not repeat the analyses with ELISA positives. When analyzing the data, please do not use different chi-squares and use GLM to fit a single model to test for geographic differences and variables related to individuals (breed, age, etc.).
There is an important questions that needs to be clarified. There exists a vaccine for West Nile virus that is used in horses. In the methods section it is indicated that vaccines only three months before blood sampling were considered. This is an important problem, you need to exclude that you have vaccinated horses in your sample. Please clarify this issue.
In the statistical section please clarify what are the dependent and independent variables you considered in your analyses. Use GLM to analyze the analyses and indicate what statistical program you used.
Results. One of the important results is the geographic differences in prevalence. It is good to include a map showing these differences. Please reduce the overlap in the analyses presented with ELISA and seroneutralization. When reporting the statistical results include the statistic you have used (chi-square or F and the degrees of freedom). You don't need to report in the tables the proportion and number of positives and negatives, please report only the prevalence (proportion of positives) and the number of samples tested.
The discussion is the section needing more work because is little focussed, please consider reordering some of the section to increase linearity. Report studies presenting humans and horses and dogs prevalences. The strong differences in the prevalence of antibodies in humans in comparison to horses is normal in this kind of studies, it is not a surprising results and reflect that humans protect ourselves from mosquito bites will horses can not do that and many of them live in very mosquito reach areas.
One important consequence of your results is that authorities need to increase surveillance and preparedness. Underdetection of human infections is an important problem associated to WNV, many infections are not detected unless doctors are aware of WNV presence in the area and familiar with the simpletons. In many cases infections are confused with dementia or other diseases in elderly people.
Lines 248-250, please provide references supporting this statement.
Lines 267-276, please review this paragraph to highlight the results of other long term studies of West Nile virus seroprevalence in horses and the variables that have been found related to antibodies prevalence, clarifying the variable and the direction of the statistical association. Please revise the logic of the structure of the discussion to made it more clear and straightforward.
Minor comments:
line 78. change to "present information on"
Author Response
Reviewer 2
Thank you so very much for your feedback and an excellent report in order to improve our manuscript. All your comments have been taken and added to our revised manuscript. Also, the manuscript has been sent to the MDPI Language Editor.
Please bear in mind that the study and samples collection have been carried out in Libya; the country that is unstable with conflict and war! We faced difficult times during our work, but with our goal to do great research we continued to go forward to our goal!
In this manuscript, the authors analyses the prevalence of West Nile virus antibodies in horses and dogs in Libya. This sort of data is necessary to understand the circulation of the virus in North Africa and improve the public health preparation for potential West Nile virus outbreaks. On the positive side of the study it is very good that the authors confirm the detection of antibodies by seroneutralization because ELISA can generate a variable number of false positives due to cross reaction with antibodies generated by other virus that do not need to be of public health concern. However, the manuscript need important improvements in the introduction, methods, results and discussion of the results.
Answer: Done. Lots of edits and corrections have been done in all manuscript sections!
In the abstract, what is important is that this study allows to determine the levels of circulation of WNV in Libya because human cases of WNV are difficult to diagnose and use to be severely underreported. See also below my comments on discussing only the results of seroneutralization tests.
Avian species are key players in the transmission of the virus and their importance and involvement in the transmission cycle is not clearly stated in the introduction where only mammals and reptiles are mentioned.
Answer: You are right! Done! Birds have been mentioned in different parts of the manuscript!
In the methods section you need to clarify how you decided if the antibodies correspond to West Nile virus or to Usutu. You mention that only West Nile virus antibodies were found but it is highly unlikely that all the seroneutralizations were negative for Usutu so I presume that some of them were positive but the titres for West Nile were higher. Usually a threshold is used of four fold differences in titre to consider them specific for one or the other virus. I also suggest that you do not repeat the analyses using ELISA positive or Seroneutralization positives, this is very repetitive and confussing for the reader. Please, consider only positives by seroneutralization as positives and do not repeat the analyses with ELISA positives. When analyzing the data, please do not use different chi-squares and use GLM to fit a single model to test for geographic differences and variables related to individuals (breed, age, etc.).
Answer: Done!
There is an important questions that needs to be clarified. There exists a vaccine for West Nile virus that is used in horses. In the methods section it is indicated that vaccines only three months before blood sampling were considered. This is an important problem, you need to exclude that you have vaccinated horses in your sample. Please clarify this issue.
Answer: The vaccination of WND is not practiced in Libya. All animals used in the study were not vaccinated! This has been stated in revised manuscript.
In the statistical section please clarify what are the dependent and independent variables you considered in your analyses. Use GLM to analyze the analyses and indicate what statistical program you used.
Answer: Statistical analysis has been added in more details in the Materials and Methods, Results and Discussion section in the manuscript!
Results. One of the important results is the geographic differences in prevalence. It is good to include a map showing these differences. Please reduce the overlap in the analyses presented with ELISA and seroneutralization. When reporting the statistical results include the statistic you have used (chi-square or F and the degrees of freedom). You don't need to report in the tables the proportion and number of positives and negatives, please report only the prevalence (proportion of positives) and the number of samples tested.
Answer: We do not think that the map of Libya in needed. Tables 3 and 4 have been removed as Table 2 would be enough!
The discussion is the section needing more work because is little focussed, please consider reordering some of the section to increase linearity. Report studies presenting humans and horses and dogs prevalences. The strong differences in the prevalence of antibodies in humans in comparison to horses is normal in this kind of studies, it is not a surprising results and reflect that humans protect ourselves from mosquito bites will horses can not do that and many of them live in very mosquito reach areas.
One important consequence of your results is that authorities need to increase surveillance and preparedness. Underdetection of human infections is an important problem associated to WNV, many infections are not detected unless doctors are aware of WNV presence in the area and familiar with the simpletons. In many cases infections are confused with dementia or other diseases in elderly people.
Answer: Done!
Lines 248-250, please provide references supporting this statement.
Answer: Done!
Lines 267-276, please review this paragraph to highlight the results of other long term studies of West Nile virus seroprevalence in horses and the variables that have been found related to antibodies prevalence, clarifying the variable and the direction of the statistical association. Please revise the logic of the structure of the discussion to made it more clear and straightforward.
Answer: Done!
Minor comments:
line 78. change to "present information on"
Answer: Done!
Round 2
Reviewer 1 Report
Comments and Suggestions for Authors
The authors responses to my queries have been consistently inadequate. I recommend that instead of providing brief responses such as "Done," it would be more constructive if the author could provide more comprehensive feedback both within the cover letter and the manuscript itself. Additionally, I would like to reiterate my previous suggestion to consider interpreting statistical values rather than relying solely on percentage values within the manuscript.
Comments on the Quality of English LanguageModerate English Editing
Author Response
The authors responses to my queries have been consistently inadequate. I recommend that instead of providing brief responses such as "Done," it would be more constructive if the author could provide more comprehensive feedback both within the cover letter and the manuscript itself. Additionally, I would like to reiterate my previous suggestion to consider interpreting statistical values rather than relying solely on percentage values within the manuscript.
Answer: Thank you for your feedback and sorry for being very brief in my reply. hence, you will find all detailed answers to your questions, and hope it will satisfy you and answer all your inquiries:
Abstract: Add statistical values, where applicable.
Answer: Significant difference was found between the WNV seroprevalence in the western and eastern areas of Libya and the other factor that increased the risk of being exposed to WNV infection was the age of animals. Also, more statistical values have been added in the main article.
Keywords: Rephrase as “West Nile Virus; Horses; Dogs; Seroprevalence; Libya”.
Answer: Done! Keywords have been rephrased as suggested by the Reviewer: “West Nile Virus; Horses; Dogs; Seroprevalence; Libya”.
Introduction:
Include details regarding the pathogenesis and the significance (clinical consequences) of West Nile Virus infection in horses and dogs.
Answer: Provided! For example:
WNV replication may occur in several organs and tissues including lymph nodes, spleen, kidney, muscle and the central nervous system [10]. Our current understanding of the disease includes neurologic and neuroinvasive disease where the virus attacks and infects the nervous system as well as other organs [11-13].
Mammals, following WNV infection, may develop low titer of viraemia after subcutaneous or intravenous inoculation with WNV (Guthrie et al., 2003). Dogs and horses can be important sentinel for WNV circulation and potential human exposure; horses as they are very susceptible to WNV infection and dogs as they can give important information on the WNV circulation in urban areas and in environment close to humans. They can develop good level of antibody titer that are easily measured [14].
Viremia is usually low and decreases significantly one to two days after symptom onset (Petersen et al., 2003). Line No. 62: “Not mammal species” Is this phrasing accurate?
Answer: In other mammal and nonmammal species including dogs, cattle, sheep, goats, camels, deer, squirrels and reptiles, WNV infection can elicit antibodies. Humans, horses and other mammals are incidental “dead-end” hosts because WNV-related vectors are mainly ornithophilic and due to the inadequate level of viremia needed for infection and transmission by mosquitoes. These animals develop titers of WNV-specific antibodies and low-level viremia following infection.
Line No. 70-71: Rephrase the sentence.
Answer: However, no information is available about WNV circulation in Libya. There is only one published study on West Nile disease (WND) seroprevalence in humans in Libya; this study was published in 2017, and it showed 11 positive samples out of 400 samples tested (2.75%) using enzyme-linked immunosorbent assays (ELISAs) [16].
Materials and Methods:
Line No. 84: Was there a specific reason for the extended three-year duration needed to complete this seroprevalence study?
Answer: Yes, the reason that we were working in unstable country “Libya” with conflict and war. It was not easy to go out and collect samples due to security reasons. Some days the fighting using weapons were in the streets!
> Why did the authors opt for collecting 574 blood samples from horses while only collecting 63 from dogs? As an epidemiological study, did the authors employ any particular epidemiological formula for determining the sample sizes?
Answer: To determine the sample size (n) in order to estimate the rate of WNV, p (D+) in the horse population, estimates (guesstimates) have been calculated depending on the probable level of the disease (p) from the available evidence of the regional and neighbouring countries. The guesstimates were set to approximate 30% of horse population will have antibodies to WNV, and also, assuming that the survey estimate to be within 4% of the level.
So: P = 0.3 Q = 1-P = 0.7 L = 0.04
n= 4PQ/L2 = 4X0.3X0.7/(0.04)2 = 525
10% was added to the sample size to reduce any non-response bias (n=574).
The proportional sampling was calculated from each selected area depending on the number of horses in the area. Number of samples in Al Zawia area were 110 (Surman, Al Zawia and Zawarah), and in Tripoli area were 365 (Tripoli, Gasr Ben Ghashir and Al Swani), and in Al Marj were 99 samples.
For dogs, the samples were collected from Tripoli only, depending on the results published on WND in humans in Libya [16], so dogs’ samples were collected from dogs living in the same area where the positive cases of human WND have been reported, assuming that dogs can be considered as good sentinels for monitoring WNV.
> Likewise, what criteria were employed for selecting samples within breeds and across various regions?
Answer: The criteria for identifying hotspot areas are:
- Suitable environment for vectors based on rainfall precipitation and green vegetation areas (FAO RVF Early Warning/Decision Support Tool).
- High animal density and movements based on animal movement survey and information available at National Center for Animal Health (NCAH) in Libya.
Due to the unavailable epidemiological data on WND in Libya, the expected prevalence was based on the available published information from the regional and neighbouring countries. The guesstimate of the prevalence was 0.3 with 95%.
> Since Table 2 encompasses the results of Tables 3 and 4, I recommend excluding Tables 3 and 4 from the article.
Answer: Accepted and Tables 3 and 4 have been removed!
> Why variables such as breed and age were presented for horses only but not for dogs? Could you please provide an explanation for this discrepancy?
Answer: For dogs, few samples were collected from Tripoli only, depending on the results of the single paper published on WND in humans in Libya, so dogs’ samples were collected from dogs living in the same area where the positive cases of human WND have been reported, assuming that dogs can be considered as good sentinels for monitoring WNV.
> Upon reviewing the Results and Discussion sections, it appears that the results were not presented based on statistical analysis. Merely discussing the results based on percentage values, without taking into account the statistical analysis, lacks scientific rigor and will not provide a meaningful contribution to the scientific community.
Answer: Statistical analysis has been added in more details in the Materials and Methods, Results and Discussion section in the manuscript!
Discussion:
> Line No. 230-233 and 233-237: Rephrase the sentences.
Answer: Done!
> In discussion section, also add explanation regarding the significant findings of the study, such as the seroprevalence in the western and eastern regions of Libya, as well as the seroprevalence observed in the higher age groups.
Answer: Different prevalence values were also found between the western and eastern regions of Libya; the western regions were more affected than, the eastern regions. One possible explanation for this is that the majority of samples collected from the eastern region were from horses of young ages.
We find out that age group was the only risk factor with stronger association from seropositive of ELISA. Even ELISA positive samples in young age less than 18 months were all negative by VNT, confirming that the disease is observed in older ages (Table 5).
Conclusion:
Conclusion is poorly written. Please provide a concise conclusion based on the study's significant findings and avoid unnecessary explanation.
Answer: Conclusion: The present study has provided novel evidence about the circulation of WNV in animal/vector populations and in certain environments of this country. This has added new knowledge to the ongoing documented endemic status of the virus in North Africa and its possible emergence as an important human health problem.
There is an urgent need for continuous monitoring programmes for humans, horses, mosquitoes and birds, including migrating avian species, to provide essential epidemiological data for the early detection of WNV circulation. It is equally crucial to increase public and professional awareness about WNV and associated clinical problems in animals and humans.
Reviewer 2 Report
Comments and Suggestions for Authors
The authors have missinterpreted by recommendation. I recommended focussing in the presentation of VNT instead of ELISA, however in the new version the statistical models are done with Elisa instead of VNT results.
Minor questions:
line 137. You mean WNV.
Author Response
The authors have missinterpreted by recommendation. I recommended focussing in the presentation of VNT instead of ELISA, however in the new version the statistical models are done with Elisa instead of VNT results.
Answer: Thank you for your feedback and sorry for the misunderstanding. lots of edits and corrections have been made following your valuable comments, for example:
To validate our results, and also to take out the effect of ELISA false positive from the results, we generate another logistic regression run by using VNT as dependent variable with the same previous risk factors, we found out that age group was the only risk factor with stronger association for positive ELISA. In addition, positive ELISA of young age group less than 18 months of age was 100% VNT negative.
Table 5 has been amended to include VNT results.
line 137. You mean WNV.
Answer: corrected!
Round 3
Reviewer 1 Report
Comments and Suggestions for Authors
Add p value in the text and tabular data, where applicable.
Comments on the Quality of English LanguageMinor English language editing
Reviewer 2 Report
Comments and Suggestions for Authors
I have seen this manuscript three times and the authors are not following the reviewers advice. In the current version they have changed some results, but not the others and results are discussed that are not presented in the results section.
Comments on the Quality of English LanguageThe revised parts needs editing